# Inflaming Public Interest: A Qualitative Study of Adult Learners’ Perceptions on Nutrition and Inflammation

**DOI:** 10.3390/nu12020345

**Published:** 2020-01-28

**Authors:** Stephanie Cowan, Surbhi Sood, Helen Truby, Aimee Dordevic, Melissa Adamski, Simone Gibson

**Affiliations:** Department of Nutrition, Dietetics and Food, Monash University, Level 1 264 Ferntree Gully Road, Notting Hill, VIC 3168, Australia

**Keywords:** nutrition education, nutrition misinformation, credibility, nutrition science

## Abstract

Research suggests national dietary guidelines are losing public resonance, with consumers actively seeking alternate nutrition advice from unregulated online platforms that often propagate misinformation. Improved diet quality can beneficially affect inflammation, and with science relating to nutrition and inflammation also appealing to consumers, this emerging topic provides an opportunity to consider how novel engagement strategies can be used to increase public support of expert-generated advice. This study aimed to qualitatively explore MOOC learners’ perceptions and experiences of following diets believed to help manage inflammation. Data were collected from an evidence-based nutrition-focused Massive Open Online Course (MOOC), which included a unit titled Foods and Inflammation. The Framework method was used to analyze 12,622 learner comments, taken from the MOOC’s online discussion forum and questionnaire. Learners identified avoidance of core food groups, such as dairy and grains, as key in managing inflammation. Dietary advice came mainly from the internet, and health professionals reportedly lacked an appreciation of the learners’ underlying nutrition knowledge, providing oversimplified advice that did not satisfy their scientific curiosity. To help build consumer trust and increase engagement, health professionals need to consider innovative education strategies that utilize novel topics such as nutrition and inflammation, in a safe and accurate manner.

## 1. Introduction

National dietary guidelines provide evidence-based recommendations to ensure optimal nutrition and prevention of chronic disease for populations [1,2]. However, research suggests that the guidelines may not resonate with all consumers [3]. A UK survey of 1751 adults found that 48% of consumers did not want the government to provide advice on what they eat [4]. Similar sentiments are shared by US consumers, with 70% stating that the government should not tell people what to eat, and 43% complaining that they were tired of hearing the government’s nutrition campaigns [5].

Consumers who are interested in nutrition science, and believe they are relatively well-informed, can be especially difficult to engage in nutrition campaigns that utilize traditional communication strategies [6]. These consumers are generally distrustful of the governments’ role in science, and believe the public should have a larger role in making decisions about key nutrition issues [4,5]. This highlights the limitations of traditional expert-driven nutrition messages, such as national dietary guidelines, which do not adequately consider the publics’ diverse attitudes towards nutrition science.

With chronic disease accounting for the majority of deaths worldwide (63%), and rates steadily increasing [7], there is a need for health professionals to look beyond national dietary guidelines, and utilize a variety of engagement strategies [8]. These strategies should include innovative nutrition messages that can garner greater support from distrustful consumers [9,10]. Reframing traditional delivery methods used to promote healthy eating, may increase public engagement with expert-driven recommendations, and in the longer-term optimize health outcomes. Since inflammation and nutrition is an emerging area of interest for scientists and consumers alike, this topic provides a useful opportunity to explore how novel engagement strategies can be used to increase public support of expert-generated advice.

Chronic low-grade inflammation, or meta-inflammation, is involved in the development and progression of a range of chronic diseases, such as type two diabetes mellitus (T2DM) and cardiovascular disease (CVD) [11,12,13]. A number of modifiers have been shown to influence inflammatory marker levels, with smoking, older age, greater body fatness, physical inactivity and poor diet quality, being associated with elevated levels of circulating pro-inflammatory markers [14]. Specific nutrients and foods, including but not limited to, omega-3 fatty acids from fatty fish [15,16], saturated fat from processed foods and animal meats [17], fiber from low glycemic index (GI) wholegrains [18,19], and carotenoids from fruits and vegetables [20], have been shown to modify acute phase proteins, cytokines and adipokines, such as C-reactive protein (CRP) [18,19,20], adiponectin [15,16], tumor necrosis factor-α (TNF-α) [15] and interleukin-6 (IL-6) [17]. This is accompanied by accumulating evidence suggesting that improved diet quality, using a whole-of-diet approach focused on implementing healthy dietary patterns, such as the Mediterranean diet, has beneficial effects on biomarkers of inflammation, including CRP and leptin [21,22,23]. Further, the Dietary Inflammatory Index (DII), a tool established in 2009, is now frequently used in research to categorize individuals’ diets on an anti- to pro-inflammatory continuum [24].

The science relating to nutrition and inflammation also appeals to consumers. With Google ^TM^ searches for ‘inflammation and diet’ and ‘inflammation and nutrition’ generating over four million results (searched on the 3rd September 2019). However, with no evidence-based dietary guidelines available that specifically address inflammation [25], healthcare professionals may be resistant to provide guidance on this emerging nutrition topic, and it is likely that the majority of available information is generated from unregulated sources. The 21st century has seen an unprecedented increase in public platforms afforded by social media, providing everybody with the power to create and share information. It is this decentralization of nutrition information that dilutes evidence-based nutrition messages, and propagates nutrition misinformation [26,27].

Growing disregard for expert-generated nutrition advice, coupled with unlimited and autonomous access to unregulated online platforms, necessitates the need for health professionals to provide dietary advice that can satisfy the scientific curiosity of all consumers in a responsible way [9,28,29,30]. Although evidence is still emerging regarding the effects of diet on inflammation, research indicates that the core principles of anti-inflammatory eating are consistent with National dietary guidelines [1,2]. These include promoting consumption of healthy fats, prioritizing low GI wholegrains, and optimizing consumption of dietary anti-oxidants through a diverse intake of fresh fruits, vegetables, and herbs and spices [31]. In this sense, when providing advice on nutrition and inflammation, it is not the recommendations that change, but rather the education strategy used to engage the consumer. The topic of nutrition and inflammation, may provide a vehicle for health professionals to responsibly disseminate novel nutrition messages, in a manner that satisfies the publics’ diverse nutrition interests, and hence may resonate with more consumers.

With this in mind, this research has used data collected from an evidence-based nutrition-focused Massive Open Online Course (MOOC) [32], to better understand adult learners’ perceptions and experiences with nutrition and inflammation. This course, titled ‘Food as Medicine’, was developed as a starting point for members of the general public interested in learning more about food, and how it can be used to support health. The first week included a unit ‘Foods and Inflammation’, due to its popularity in the public sphere. Data were gathered from the online discussion forum embedded within the MOOC, where learners freely engaged in natural dialogue. This type of data is known as Real World Data (RWD), and is increasingly recognized as useful in research to better understand consumers perspectives and experiences [33]. It may also provide a real-time representation of consumers actively seeking nutrition information online. To date, more than 150,000 people from 158 countries registered for this course, highlighting the popularity of these types of novel nutrition topics.

The aims of this paper were to: (1) explore MOOC learners’ perceptions of nutrition and its effects on inflammation, and (2) explore MOOC learners’ experiences of following different diets believed to help reduce inflammation. Such exploration will help health professionals understand how to responsibly tailor traditional evidence-based nutrition messages, so that they can better meet the diverse needs and interests of all consumers. 

## 2. Materials and Methods 

The MOOC was designed to address contemporary issues in nutrition and was targeted for the general public. Overall, the course emphasized a whole food and dietary pattern approach, and aimed to promote healthy food and eating practices. Registration for the MOOC opened on the FutureLearn platform (The Open University, Milton Keynes, UK) to a worldwide audience and was available to anyone with internet access. No prior knowledge of nutrition, health or science was assumed. 

Foods and Inflammation featured in week one of the three-week course. It was one of 13 core topics covered by the MOOC. Learners were encouraged to read articles, watch videos and discuss their experiences and knowledge of nutrition and inflammation. 

The MOOC was developed by academics and health professionals including dietitians, nutritionists and medical professionals in the Department of Nutrition and Dietetics at Monash University (Melbourne, Australia). 

As this is a qualitative study, the Qualitative Research Review Guidelines (RATS) for reporting qualitative data was followed, and the complete RATS checklist can be found in Appendix A. 

Data collection - The MOOC is open access and data collected from July 2017 to July 2018 was included in the analysis. Quantitative and qualitative data were collected via an online questionnaire, and further qualitative data was collected from online forum posts. The questionnaire was developed by the researchers and tested for usability and face validity (Appendix B). The questionnaire contained 25 questions. Quantitative data collected via the questionnaire included age, gender, inflammatory disease status, and where they accessed inflammation and nutrition advice. The questionnaire also sought qualitative responses using open-ended questions related to learners’ perceptions and experiences with food and inflammation (Appendix C). Questionnaire responses were collected using Qualtrics® Research Suite (Qualtrics^®^, Sydney, Australia). Further qualitative data regarding learners’ views and experiences of nutrition and inflammation were collected from the online discussion forum, using prompting questions such as ‘Within the comments, consider sharing your thoughts on the role you think diet plays in inflammation’. All qualitative data from the forum comments and questionnaire responses were downloaded into a .csv file. 

Completing the questionnaire and participating in the discussion forum were voluntary. Ethics approval was obtained by the Monash University Human Research and Ethics (CF16/905 – 2016000470), and all participants provided written informed consent. 

During the data collection period 33,508 learners completed the MOOC. Researchers analyzed 2344 comments from the discussion forum embedded within the unit ‘Foods and Inflammation’. The online questionnaire was completed by 3426 respondents, generating 10,278 open-ended question which were also analyzed.

Data Analysis - Quantitative data analyzes were conducted using Excel [34] software. Frequencies (*n*) and proportions (%) were used to measure categorical data. Qualitative data taken from the online discussion forum and the questionnaire, were combined and analyzed as one dataset. Analyses were undertaken using the framework method [35]. This method provides clear steps to follow and produces highly structured outputs of summarized data. The framework method was ideal for managing the large dataset, which incorporated 12,622 data points from thousands of participants. It helped generate a holistic overview of the entire dataset via grouping the large number of participants into suitable cases.

### 2.1. Familiarization and Coding

Three researchers (SC, SS, SG) familiarized themselves with the data. Two researchers (SC, SS) used open coding independently for 200 forum comments, and came together to discuss the codes that had been assigned to each comment. 

### 2.2. Developing A Working Analytical Framework 

Working through all comments, two researchers (SC, SS) discussed why it had been interpreted as meaningful, what it revealed about participants views on diet and inflammation, and how it might be useful in answering the research questions (Appendix D). After discussion, researchers agreed on a set of codes, each with a brief description, forming the initial analytical framework. The same two researchers then independently coded further comments using the initial framework, taking care to note any new codes under ‘other’ to avoid ignoring data that did not fit the existing set. The researchers then met again to revise the initial framework to incorporate new refined codes and to group conceptually related codes together. This process of refining, applying and refining the analytical framework was repeated throughout the entirety of the coding process. The final framework consisted of 200 codes, clustered into 13 categories, each with a brief explanatory description of their meaning, examples of what elements might be summarized under that code, and an illustrative quote (Appendix D).

### 2.3. Applying the Analytical Framework 

Two researchers (SC, SG) applied the analytical framework to all comments using NVivo [36] qualitative data analysis software, systematically highlighting each comment from the online discussions and open-ended questions in the questionnaire, and assigning an appropriate code from the analytical framework. 

### 2.4. Charting Data into the Framework Matrix

Once all the data had been coded using the analytical framework, researchers summarized the data into 18 matrices using NVivo [36] software. The matrices comprised of one row per MOOC run (e.g. run 4, 5, 6 and 7) and questionnaire questions (e.g. question 7, 10 and 12) and one column per code. A separate matrix was used for each category. The verbatim words that were assigned to each code during the coding process were inserted into the corresponding cell in the matrix.

### 2.5. Interpreting the Data 

Themes were generated from the dataset by reviewing the matrix and making connections within and between MOOC runs and categories. During the interpretation stage, the research team tried to go beyond descriptions of each category towards developing themes which offered possible explanations for what was happening with the data. Figure 1 provides a visualization of the flow of data from coding through to interpretation. It does so via mapping the interactions between cases, codes, categories and themes.

### 2.6. The Research Team

Three authors were Accredited Practicing Dietitians (SC, HT, SG) with expertise in nutrition and inflammation, online learning and educational research. One author was a Registered Nutritionist (nutrition science) (AD) with expertise in inflammation. One author was a tertiary student (SS) completing her final year of an undergraduate nutrition degree. 

## 3. Results

### 3.1. Learners’ Backgrounds

Eighty-eight percent of questionnaire respondents were female, 56% were aged ≥ 18 and ≤ 55 years, and 40% were ≥ 56 years. Forty-four percent stated that they had previously followed an anti-inflammatory diet to prevent or help manage inflammation. The majority of questionnaire respondents sourced information on diet and inflammation from the internet (38%). Table 1 summarizes the sources of nutrition and inflammation content reported by questionnaire respondents, when asked to identify the top three sources of information used to help implement recommendations for anti-inflammatory eating. Table 2 summarizes the diseases and conditions reported by questionnaire respondents, when asked whether they had an inflammatory condition. Their answers included not only autoimmune or immune mediated conditions, but also lifestyle and mental health diseases such CVD and depression. Both questions were open ended and participants could include any information they deemed relevant.

### 3.2. Learners’ Perceptions and Experiences 

From the analysis of the discussion forum and responses to open-ended questions six themes were identified, each containing a number of categories. Table 3 summarizes each theme, listing the categories and key illustrative quotes.

#### 3.2.1. A range of Nutritional Factors Influence Inflammation

Learners reported a wide range of dietary factors as having anti-inflammatory and pro-inflammatory effects, ranging from diets, to foods and nutrients (Figure 2).

Learners commonly referred to the Mediterranean diet and plant-based diets as being anti-inflammatory. However, there was a large range of diets reported, including the low FODMAP diet, paleo, ketogenic, and the “Eat Right for you Blood Diet”. 

Foods purported to be anti-inflammatory included vegetables, fruits, nuts and seeds, herbs and spices, olive oil, oily fish, legumes/lentils and wholegrains. However, learners also emphasized the importance of eliminating whole food groups. While these included discretionary food items such as processed foods, soft drink and alcohol, they also included core food groups, specifically dairy, red meat and grains/cereals (especially bread and pasta). 

A number of diet types and foods caused contention between learners, with some reporting them as pro-inflammatory and others as anti-inflammatory. For example, while the Mediterranean diet was advocated as being anti-inflammatory, common Mediterranean foods such as tomatoes were often perceived as pro-inflammatory due to being a “nightshade” or “acidic”. Other foods commonly reported to be both pro- and anti-inflammatory included citrus fruits, spicy foods, soy, corn, rice, legumes, peanuts, eggs and fish.

Nutrients thought to provide anti-inflammatory effects included omega-3 fats, fiber and vitamin C. Sugar, gluten, salt, caffeine, lactose and artificial sweeteners were popularly thought of as pro-inflammatory. 

Learners also placed an emphasis on the importance of nutraceuticals, often supplementing their diets with curcuminoids, fish oil, pre- and pro-biotics, glucosamine and magnesium. Less common products included Taurine, 5-HTP, collagen powder and Dopa Mucuna.

#### 3.2.2. Managing Inflammation is More than Just Diet

Learners voiced the need for using a holistic approach to managing inflammation, and stressed the importance of including regular physical activity, maintaining a healthy weight, and looking after their mental health, using strategies such as mindfulness and meditation for stress management. 

In addition to these more traditional lifestyle factors known to affect health, learners also stressed the importance of considering more contemporary topics. These included maintaining a healthy “gut microbiome”, and utilizing sustainable environmental practices, including eating seasonal, free-range and organic produce.

#### 3.2.3. Anti-Inflammatory Eating Transformed My Health

Many learners reported that their anti-inflammatory ways of eating had extremely positive effects on their health (Figure 3). Some learners reported immense benefits from their diets, such as curing their disease or leading to its remission. This was especially true for diabetes, heart disease, and functional gut disorders such as irritable bowel syndrome. Though dietary benefits were also attributed to autoimmune and immune-mediated conditions, including rheumatoid arthritis, multiple sclerosis, asthma and ulcerative colitis.

Objective evidence was used to provide proof of the benefits attributed to dietary changes, including reporting measurements of lowered of blood pressure, cholesterol, glucose and weight. Learners stated that they reduced their medication, and listed symptoms that had improved following dietary change, including aches/pains, mobility, bloating and headaches.

Perceived benefits also related to general health and wellbeing, with many learners using anti-inflammatory diets in the absence of disease as a preventative measure. These improvements related more to subjective measures, using terms such as vitality, mood, clarity of thoughts and sleep quality.

#### 3.2.4. I Need Guidance but Who Do I Trust

Learners frequently reported being overwhelmed by the large amount of information available, and were frustrated with misinformation and conflicting dietary advice, mostly relating to information sourced from the internet. Many recognized that they lacked skills required to filter and interpret the abundance of available information, and reported that many of the diets were complicated and difficult to follow. Learners also frequently distrusted food companies and perceived their marketing strategies as deceptive, making it hard to choose healthy foods. 

Overwhelmingly, learners were frustrated that they did not receive what they perceived to be adequate dietary advice from healthcare professionals. They discussed doctors promoting medications over dietary counselling when managing inflammatory conditions, and health professionals focusing on treating symptoms rather than the underlying causes of inflammation. Learners described their requests for more information on lifestyle management strategies as being minimized, rebuffed or ignored. 

Qualified healthcare professionals were largely seen as being averse to contemporary nutrition advances. Specifically, nutrition experts were viewed as being resistant towards providing dietary advice, unless the suggested management strategies were well supported by evidence, which learners often perceived as outdated. Where dietary advice was provided, it was described as being either too vague for learners to confidently implement dietary change, or as being given in an authoritative manner, with no compromise or room to alter advice according to learner needs. 

#### 3.2.5. Why it’s so Hard to Follow an Anti-Inflammatory Diet 

High cost and poor availability of foods, often related to the use of specialty products (e.g. gluten free or organic), were identified as important limiting factors when implementing anti-inflammatory diets. 

Some barriers arose because dietary advice conflicted with learners’ social obligations. Where learners were responsible for the household cooking, it was not feasible for the entire family to also follow the diet. Learners also struggled with social restrictions imposed by the diet, limiting their ability to eat out. Hence, the additional meal planning and cooking required to sustain the diet was seen as being time consuming and inconvenient. 

Some learners reported increased lethargy, persistent hunger, and diminished enjoyment from eating, all of which were related to the restrictive and repetitive nature of the anti-inflammatory diets they followed. Learners also stated that when they focused on food avoidance, they felt a strong sense of deprivation and experienced food cravings. 

#### 3.2.6. What I need from Health Professionals

Learners identified that deficits in their nutrition knowledge limited their ability to implement dietary change, thus they requested a range of learning tools, many of which directly related to the aforementioned barriers. For example, learners identified the need for meal planning and shopping tips to lessen time burdens and improve efficiency, and the provision of recipes to increase sensory enjoyment and ensure nutritional adequacy. 

Learners’ also identified the need for structured food lists, label reading resources, information around meal timing and portion sizes, and education regarding why dietary changes need to be implemented (not just how). Collectively these requests for further nutrition education provide insights into the high level of detail learners would like nutrition professionals to offer.

Learners expressed difficulty waiting for long-term gains. They emphasized the importance of using small and progressive goals, and having at least one measurable or observable benefit to motivate them in maintaining dietary change. These benefits were not reserved to traditional measures, such as anthropometric and biochemical markers, but also included subjective measures such as fatigue, sleep, mood, changes in joint stiffness and/or mobility. 

Personalized nutrition, rather than a “one size fits all” approach, was repeatedly recognized by learners as key to effectively managing their health. 

## 4. Discussion

In this study, learners of a nutrition-focused massive open online course (MOOC) shared their perceptions of nutrition and inflammation, as well as their experiences with anti-inflammatory eating. The focus on anti-inflammatory diets was chosen due to its consumer appeal. This trending nutrition topic provided a useful exploration of how reframing traditional nutrition messages, using a novel lens, may help to improve the publics’ engagement with expert-driven nutrition recommendations. 

Learners were following a number of different lifestyle strategies deemed helpful in preventing or managing inflammation. The majority of information sources on this topic came from the internet, and learners largely viewed health professionals as being averse to providing advice on diet and inflammation. The few learners who did receive advice from qualified health professionals, described recommendations as lacking complexity, and failing to capture their interests. To generate greater engagement in expert-generated nutrition advice, it is important that we consider how these findings impact future nutrition education strategies. 

Overall MOOC learners displayed a sophisticated understanding of key modifiers of inflammatory markers, although some beliefs were contradictory and inconsistent with the current evidence. The broad range of diets, foods, and supplements being used by learners to help manage or prevent inflammation, highlights the vast and diverse nature of information available to consumers. While some of the diets being followed had an established evidence base, such as the Mediterranean or DASH diets, many learners identified avoidance of core food groups as key in managing inflammation. This is especially concerning when you consider that National dietary guidelines emphasize the importance of variety, both across and within food groups, to ensure the diet contains sufficient amounts of nutrients essential for health [1,2]. Findings from this study indicate that the nature of dietary advice available to MOOC learners, may be compromising the nutritional adequacy of their diets. 

Learners’ poor level of engagement with nutrition professionals, and propensity to access alternate nutrition advice online is consistent with available literature. Earlier research has reported that 78% of Australian adults use the internet to find health-related information [37], and that internet sites such as YouTube and Facebook are the fasted growing sources of health advice [38]. Notably, 24% of MOOC learners sort guidance on nutrition and inflammation to help manage poorly understood inflammatory conditions, specifically fibromyalgia, chronic fatigue syndrome and rheumatoid arthritis. These conditions lack clear guidelines for symptom management [39,40], and sufferers are more likely to seek health information from social networking sites [41], and report high rates of alternative therapy use [42]. To quell the publics’ engagement with unregulated sources of nutrition misinformation, health professionals must learn how to responsibly communicate dietary guidance in the area of inflammation, especially in high-risk patients with complex inflammatory conditions. 

Nutrition experts possess the skills and knowledge required to provide safe and accurate dietary advice on emerging nutrition topics that attract public interest, such as nutrition and inflammation. They are also uniquely qualified to actively correct food and nutrition misinformation [43]. However, to do so effectively, consumers need to feel supported and have trust in their healthcare providers [10,44]. This was not the experience of learners in the current study, who described feeling ignored and criticized by health professionals when seeking guidance on nutrition and inflammation. While learners perceived nutrition professionals to readily discredit nutrition misinformation, in doing so they also compromised the provision of patient-centered care [45], with learners feeling left out of the decision-making process, and voicing that the care provided did not meet their individual needs. The reluctance of healthcare professionals to provide nutrition recommendations in the absence of a strong evidence base creates a quandary. How do health professionals provide safe and balanced nutrition advice that also meets the needs of consumers who want to try novel approaches to eating? 

Applying the constructivist learning theory [46] may assist health professionals to better engage the publics’ interest in expert-generated nutrition messages, not only during one-on-one consultations, but also when delivering online nutrition content. According to this learning theory, nutrition experts must be: a) concerned with the experiences and contexts that make the learner willing to learn; and b) facilitate extrapolation, where the learner constructs new knowledge based upon current understanding. By allowing the learner to construct their own learning, nutrition experts will grasp a better understanding of the motivations for learning; and tailor their advice to suit the unique needs of the learner, so that the information provided builds upon their existing knowledge base and experience, rather than discrediting it [47]. 

Learners fervently believed that following an anti-inflammatory diet provided a spectrum of benefits, ranging from curing inflammatory conditions such as rheumatoid arthritis, to improving depression, and reducing risk markers in metabolic disease. They were also skeptical of health professionals who disregarded the validity of their experiences. These findings suggest that questioning the relevance of the learners’ motivations, may cause them to become disengaged and seek information from other potentially misleading sources. Nutrition professionals should not only base their management on available evidence, but also appreciate the patients lived experience. In the 21^st^ century, where the public can source unregulated health information at the click of a button, a more open-minded approach may help nutrition professionals to maintain public interest and trust [8]. 

When encouraged to provide information on the challenges faced when implementing anti-inflammatory diets, learners eagerly shared their knowledge deficits and actively sort clarification. They also revealed a preference for the delivery of detailed information, including guidance on food pairing and recipe adaption, and emphasized the desire for more complex scientific explanations of the nutritive mechanisms responsible for improved health outcomes. This illustrates the importance of encouraging learners to guide the learning process, so that the content being delivered builds upon their current level of understanding, and hence captures their curiosity. It also suggests that health professionals may be underestimating the publics’ enthusiasm towards nutrition science, and their ability to understand and synthesize nutrition information. 

While learners in this study discussed being frustrated by nutrition misinformation, they were relatively well informed on the topic. Not only were they aware of the role that chronic low-grade inflammation plays in disease development and progression, they also identified emerging research that suggests inflammatory modifiers, including stress and large bowel microbiota, may play a role in regulating circulating levels of inflammatory markers [14]. These observations suggest that for this motivated and well-informed sub-set of consumers at least, health literacy skills may be underrated. This is consistent with recent data (2018) from the Australian Health Literacy Survey (HLS), showing that the majority of survey respondents (*n* = 21,315) were confident in their level of health literacy when finding, appraising and understanding health information [48]. While earlier data (2006) from the Australian Adult Literacy and Life Skills Survey (ALLS) showed that just under half (41%) of survey respondents (*n* = 8988) were assessed as having adequate health literacy skills, it also highlighted that health literacy was higher in people who had completed a Bachelor degree (70%) and among people with high incomes (63%) [49,50]. This reinforces that nutrition education should not always be short, simple and without nutrition jargon [43], but rather should match the learners’ current levels of understanding.

The strengths of this research include the large international sample size and the breadth of responses. The broad application of these findings should not be underestimated, with RWD gathered from thousands of learners with diverse motivations for seeking advice on anti-inflammatory eating. Some learners were apparently healthy and sort advice for the prevention of inflammation, others reported health conditions that had no overt inflammatory pathophysiology, such as irritable bowel syndrome and depression, while others sought guidance to help manage more complex inflammatory conditions. These findings also highlight recommendations that are relevant not only to the provision of information on nutrition and inflammation, but may be applied to other nutrition topics with emerging scientific evidence. Limitations include the likelihood that a university developed nutrition-focused MOOC attracted health-minded participants who are highly engaged with information on nutrition science. However, we believe that this further demonstrates the importance of tailoring traditional nutrition messages, to suit the publics’ diverse needs when disseminating information. Learner demographics revealed that older women displayed the highest enrollment rates. Menopause is the dominant issue for women aged between 45–59 years, peaking in the 50–54 year age group, and hence it is possible that the findings of the current study largely represent the perceptions and experiences of menopausal and post-menopausal women. Possible explanations for their increased level of engagement include: increased risk of chronic disease associated with hormonal changes and aging [51]; longer life expectancy when compared to males, with females comprising 61% of adults aged over 80 years [52,53]; and gender differences in health care-seeking behavior [54]. 

## 5. Conclusions

This study has recognized a sub-set of the public who are disengaged with traditional health and nutrition messages, and are actively seeking out information on diet and inflammation. In the internet age where unregulated nutrition information can be freely created and shared, there is a risk that consumer interest in these types of novel nutrition topics, increases their susceptibility to nutrition misinformation. While providing safe and balanced nutrition advice, and correcting nutrition misinformation is paramount, health professionals cannot ignore that certain sub-sets of the public are now more well informed about nutrition than ever before. Inflammation and nutrition is an emerging area of nutrition research that has sparked laypersons scientific curiosity. It provides a vehicle that can be utilized by nutrition professionals to responsibly engage the public in expert-driven nutrition content. Future research should consider how health professionals can better capitalize on the learners’ past experiences without judgment, and welcome self-directed learning via discussion and taking a mentoring role. This approach will help nutrition professionals to build consumer trust, and to better tailor traditional health and nutrition messages to suit the unique needs of all consumers.

## Figures and Tables

**Figure 1 nutrients-12-00345-f001:**
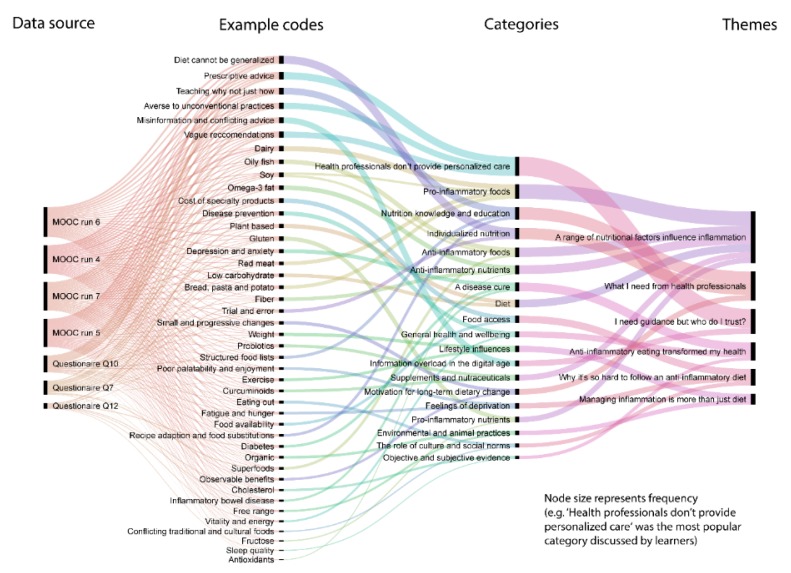
Sankey diagram visualization of the flow of data from coding through to interpretation.

**Figure 2 nutrients-12-00345-f002:**
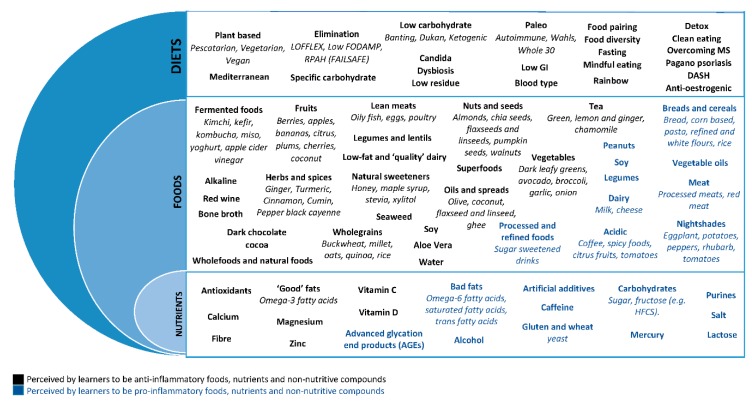
Summary of the diets, foods and nutrients reported by learners to affect inflammation.

**Figure 3 nutrients-12-00345-f003:**
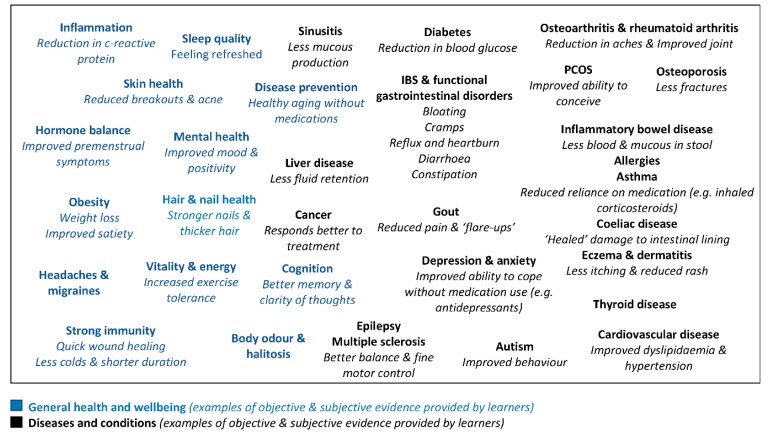
Summary of perceived benefits to following an anti-inflammatory diet reported by learners.

**Table 1 nutrients-12-00345-t001:** Sources of information used to implement anti-inflammatory diets reported by learners ^**1**^.

Sources of Nutrition Information	Number (percentage) of Questionnaire Respondents ^2^
Internet (includes search engines such as Google and social media sites)	1308 (38)
Doctors	445 (13)
Alternative health providers (includes naturopath, homeopath, chiropractor, acupuncturist, yoga instructor, personal trainer)	445 (13)
Books	411 (12)
Friends and family	343 (10)
Nutritionists	308 (9)
Dietitians	206 (6)

**^1^** A total of 3426 respondent answers are included in these summary statistics**; ^2^**respondents were asked to pick the top three answers.

**Table 2 nutrients-12-00345-t002:** Inflammatory conditions reported by learners **^1^**.

Inflammatory Condition	Number (percentage) of Questionnaire Respondents ^2^
Irritable bowel syndrome or food intolerance	994 (29)
Eczema or psoriasis	514 (15)
Depression	478 (14)
Asthma	478 (14)
Hypertension	411 (12)
Obesity	411 (12)
Rheumatoid arthritis	377 (11)
Food allergy (medically diagnosed)	308 (9)
Osteoporosis	274 (8)
Hypercholesterolemia	274 (8)
Chronic fatigue syndrome	240 (7)
Fibromyalgia	206 (6)
Inflammatory bowel disease	172 (5)
Cancer	137 (4)
Type two diabetes mellitus	133 (4)
Cardiovascular disease	133 (4)
Coeliac disease	130 (4)
Metabolic Syndrome	69 (2)
Lupus	35 (1)
Other (*n* = 66), including but not limited to endometriosis, poly-cystic ovarian syndrome, thyroid disease such as Grave’s disease, multiple sclerosis, Parkinson’s disease, chronic back pain and chronic kidney disease	1268 (37)

**^1^** A total of 3426 respondent answers are included in these summary statistics; **^2^** there was no limit to the number of conditions respondents could select.

**Table 3 nutrients-12-00345-t003:** Theme summaries, categories and key illustrative quotes generated using the Framework method of qualitative data analysis.

Theme	Categories	Illustrative Quote(s)
1.0 A range of nutritional factors influence inflammation	1.01 Diets 1.02 Foods 1.03 Nutrients 1.04 Supplements and nutraceuticals	“I went Paleo after being diagnosed with an inflammatory condition. It has truly helped.” (learner ID 58.96.80.106) “I believe a plant based diet prevents inflammatory diseases/conditions.” (learner ID 92.25.35.22) “Choosing foods with anti-inflammatory properties and foods which are more alkalising than acidic will definitely play a large role in promoting healing.” “Pro-inflammatory foods are sugar and alcohol.” (learner ID 187.201.48.75)
2.0 Managing inflammation is more than just diet	2.01 Lifestyle influences 2.02 Environmental and animal practices	“chronic high levels of stress mean that inflammation is not resolved withing the body.” (learner ID 103.85.106.87) “I don’t trust industrial foods and try to buy organic and from farmers directly.” (learner ID 202.133.214.88)
3.0 Anti-inflammatory eating transformed my health	3.01 A disease cure 3.02 Objective and subjective evidence 3.03 General health and wellbeing	“I was not expected to survive the cancer which was stage 3 when it was found but after surgery, chemo and radiotherapy and a very clean diet for a year I am still here 12 years later. The medics were very surprised.” (learner ID 31.18.251.69) “cholesterol numbers, without medication as I reacted to all tried, dropped from 9.5 to 6.5.” (learner ID 1.143.153.148) “Cannot pin point physical benefits except more energy, general feeling of wellbeing” (learner ID 5.52.109.5)
4.0 I need guidance but who do I trust?	4.01 Information overload in the digital age 4.02 Health professionals don’t provide personalized care	“There’s so much misinformation it’s hard to decipher.” (learner ID 89.120.155.144) “I have researched many books on RA and diet, most of which were difficult to follow” (learner ID 24.30.74.17) “I have not had advice on this from my GP or the dietitian I went to see, and find conflicting views online.” (learner ID 210.1.90.94) “Look for another GP. Get someone who listens to you, respects you and is willing to offer advice, not necessarily pills.” (learner ID 51.39.225.80)
5.0 Why it’s so hard to follow an anti-inflammatory diet	5.01 Food access 5.02 The role of culture and social norms 5.03 Feelings of deprivation	“It was lonely eating differently from my family and I had little energy to cook.” (learner ID 125.236.138.13) “The cost of the organic foods that I have switched to is much higher than the cheap processed foods I used to buy.” (learner ID 24.108.0.208) “Because it takes time to buy fresh food and to cook it would be easier to eat fast food” (learner ID 2.218.59.59) “The challenge however is being able to have these foods on a constant and affordable basis… because in my country we have a very small choice of special products.” (learner ID 121.214.109.127 – questionnaire)
6.0 What I need from health professionals	6.01 Nutrition knowledge and education 6.02 Motivation for long-term dietary change 6.03 Individualized nutrition	“It means different foods react in different ways with different people. Each person has to discover his/her own” through “trial and error” (learner ID 14.201.127.194 – discussion forum) “Individual advice is key” (learner ID 27.252.219.51) “Time of day for foods was very useful, so to was advice on how to pair foods to create balance, the more detail the better for me.” (learner ID 141.168.83.49) “CAN YOU GIVE ME A LIST OF ANTI-INFLAMMATORY FOODS? THANK YOU” (learner ID 51.6.145.23) “Found this helpful in knowing in such detail, why I should avoid all those tempting food ’baddies’. Thank you. Learning the science is fun!” (learner ID 74.222.74.127)

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
