# Peer review of "Inflaming Public Interest: A Qualitative Study of Adult Learners’ Perceptions on Nutrition and Inflammation"

_nutrients, 2020, doi:10.3390/nu12020345_

Round 1

Reviewer 1 Report

In this manuscript with title; Inflaming Public Interest: A Qualitative Study of Adult Learners’ Perceptions on Nutrition and Inflammation  with the aim to qualitatively explore MOOC learners’ perceptions and experiences of following diets believed to help manage inflammation. It’s a well written paper and well analyzed based on the large international sample size and the breadth of responses. The findings are novel and interesting to the readers.

To improve this manuscript, the authors should address the following suggested comments.

Major Comments

I expected the authors to clearly represent ‘Charting data into the framework matrix’ section into more interactive visual diagram like Sankey or Chord diagram. With this, it will be possible to show interaction between the nodes. It was interesting findings that Eighty-eight (88%) percent of questionnaire respondents were female, 56% were aged ≥ 18 and ≤ 55 years, and 40% were ≥ 56 years. In the discussion section, the authors should briefly what might be the cause or hypothesize.

Minor comments

In Figure 2. ‘Summary of perceived benefits to following an anti-inflammatory diet reported by learners’. The summary should be arranged according to its category but not mixed.

Reviewer 2 Report

In this manuscript, authors explore an important topic in the nutrition field: people's perceptions of nutrition and its effects on inflammation. Authors choose the relation between nutrition and inflammation because 1) this is an emerging area of interest and 2) inflammation is involved in the development and progression of a range of chronic diseases. In order to analyze this relation, authors use data collected from an evidence-based nutrition focused massive open online course that has envolved 33508 learners. Authors underline 1) the vast and diverse nature of information available to consumers to manage and prevent inflammation, 2) the propensity to use internet to find health-related information.

The paper is well written and easy to read and understand. Some clarifications are needed:

Please report in the tables the total number of people responding to the questions. Please explain better the tables 1 and 2 in the text.
